# Genome-Edited Plants: Opportunities and Challenges for an Anticipatory Detection and Identification Framework

**DOI:** 10.3390/foods10020430

**Published:** 2021-02-16

**Authors:** Alexandra Ribarits, Michael Eckerstorfer, Samson Simon, Walter Stepanek

**Affiliations:** 1Austrian Agency for Health and Food Safety, Spargelfeldstraße 191, 1220 Vienna, Austria; walter.stepanek@ages.at; 2Environment Agency Austria, Spittelauer Lände 5, 1090 Vienna, Austria; michael.eckerstorfer@umweltbundesamt.at; 3Federal Agency for Nature Conservation, Konstantinstraße 110, 53179 Bonn, Germany; samson.simon@bfn.de

**Keywords:** GMO, genome editing, regulation, detection, identification, labeling

## Abstract

It is difficult to trace and identify genome-edited food and feed products if relevant information is not made available to competent authorities. This results in major challenges, as genetically modified organism (GMO) regulatory frameworks for food and feed that apply to countries such as the member states of the European Union (EU) require enforcement based on detection. An international anticipatory detection and identification framework for voluntary collaboration and collation of disclosed information on genome-edited plants could be a valuable tool to address these challenges caused by data gaps. Scrutinizing different information sources and establishing a level of information that is sufficient to unambiguously conclude on the application of genome editing in the plant breeding process can support the identification of genome-edited products by complementing the results of analytical detection. International coordination to set up an appropriate state-of-the-art database is recommended to overcome the difficulty caused by the non-harmonized bio-safety regulation requirements of genome-edited food and feed products in various countries. This approach helps to avoid trade disruptions and to facilitate GMO/non-GMO labeling schemes. Implementation of the legal requirements for genome-edited food and feed products in the EU and elsewhere would substantially benefit from such an anticipatory framework.

## 1. Introduction

Genome editing comprises a number of different technical approaches that aim to create precise alterations in plant genomes (see [1] for a comprehensive overview). The scope of these alterations ranges from single nucleotide variants (SNVs) or small insertions or deletions introduced at a predefined position either randomly or by using a repair template, to larger sized insertions of recombinant sequence elements. Concerning detection and identification, SNVs are the most challenging type of genome edits [2,3].

In the European Union (EU), plants produced by genome editing are considered genetically modified organisms (GMOs) [4]. Genome-edited plants are thus covered by Directive 2001/18/EC [5]. In accordance with the EU regulatory framework, genetically modified (GM) plants, as well as any food and feed products derived from them [6], need to be authorized prior to their intended use, and applicants are obliged to submit an analytical method for detection and identification (Commission Implementing Regulation (EU) No. 503/2013; [7]). Besides the risk assessment and authorization procedure, the EU regulatory framework foresees rules for labeling GMOs and products thereof, their traceability at all stages of the supply chain, and plans for post-market environmental monitoring (PMEM) [6].

At the global level, most countries have introduced biosafety laws that require premarket authorization for regulated products. Unauthorized genetically modified products covered by these regulations are subject to enforcement [8,9]. Consequently, analytical detection of such products and their identification are crucial elements to implement the regulatory frameworks for GMOs, particularly for the enforcement of sampling and testing food, feed and seed products [1,10]. Legislations that comprise further regulatory requirements in addition to mandatory authorization and risk assessment, such as the labeling, traceability and monitoring of genome-edited products, face comparable challenges to the EU concerning the implementation of these requirements [8]. Detection, identification and quantification are crucial in that regard.

Analytical verification is also important for compliance with non-GMO standards [11]. Identity preservation (IP) schemes that are adopted voluntarily by producers and/or distributors such as the IP control systems in the GMO-free value chain, can only be implemented when GMO products and GMO admixtures are identifiable and quantifiable. This is especially important for the quality control of GMO-free food and feed products operated by private institutions or under national certification schemes to support freedom of choice for consumers (see [11] for an overview of such systems). The developers of GM plants have established stewardship programs to facilitate compliance with existing regulatory requirements. The biotechnology industry members on a global scale coordinate such stewardship programs. They include inter alia measures to support the operation of IP schemes [12].

Control and management measures of GMOs, including labeling, depend on reliable analytical results. As these results may affect the marketing of food, feed and seed, their accuracy is crucial for authorities, traders, farmers and consumers [13]. Detection and identification methods are furthermore relevant for certain aspects of the PMEM of authorized GMOs, such as monitoring of environmental exposure, and to establish a link between a specific adverse environmental effect and a certain GMO [14].

Current developments in modern plant biotechnology present a number of challenges related to detection and, in particular, identification of GMOs [10,15]. More and more frequently, such GMOs do not contain standard screening elements, i.e., specific genetic elements of non-plant origin that are commonly present in transgenic GMOs. Products without screening elements can, therefore, only be detected using event-specific methods. Similarly, current screening approaches would not track down genome-edited plants, as they are not expected to contain foreign DNA or screening elements. At the target site, genome editing often leads to SNVs, or deletion or insertion of only one single nucleotide [16]. Although small changes in the genome can be detected by molecular analyses, identification of genome-edited plants harboring SNVs or small genomic modifications is challenging [2]. Even though it is expected that genome editing is precise and results in plants that carry only the intended mutation, unintended mutations may occur at varying frequencies, which might be harnessed for detection. This is especially relevant for mutations occurring in close proximity to the intended mutation.

To date, no genome-edited plant has been subject to the EU regulatory authorization procedure. Thus, validated detection methods for genome-edited plants are neither available to the EU Reference Laboratory for GM Food and Feed nor to national reference laboratories for GMO control in the EU member states. Similarly, information required to implement or develop a detection method suitable for product identification may not be readily available to EU authorities.

This article explores the idea of an anticipatory framework to ensure the detection and identification of genome-edited plants and their products, particularly food and feed products derived from those plants. Such a framework would be based on available information from different sources that may be scrutinized to identify and describe genome-edited plants on the international market. In combination with a sequence-specific detection method, this information could be used to enable the identification of sequence alterations as genome-edited. Such an approach would be useful for all legislations imposing regulatory requirements for genome-edited plants and thus would strongly benefit from a coordinated international collaboration, e.g., under the umbrella of the Convention for Biological Diversity (CBD).

## 2. Analytical Detection and Identification of Genome-Edited Products

The development of a validated detection method is a prerequisite for the successful identification of a genome-edited plant or product by analytical means. Detection demonstrates the presence of a particular genetic or phenotypic modification by analytical means [1,2]. On the contrary, identification may be defined as the allocation to a specific plant or product and to a specific developer typically by event-specific detection.

The detection of a randomly introduced transgenic sequence usually provides sufficient evidence for identification of a GMO, including in food or feed products. Screening approaches for standard transgenic sequence elements and event-specific methods are commonly used to detect, and—in the case of event-specificity—identify a GM plant. Event-specific polymerase chain reaction (PCR) methods take advantage of unique sequences at the junction between inserted transgenes and plant genomic sequences. Thus, classic GMOs can be identified and attributed to specific events and developers. Event-specific methods are also useful to quantify GMOs in products such as food, feed and seed by means of quantitative PCR tests. Comprehensive GMO databases are currently available and used by accredited testing laboratories for detection and monitoring purposes for GM plants. For national reference laboratories designated in the EU under Regulation (EU) 2017/625 [17], the Joint Research Centre of the European Commission is such an information platform offering detailed sequencing data on GM plants. A comprehensive collection of validated identification methods is also provided. Additionally, there are global databases such as Biosafety Clearing House (BCH) [18], established under the Cartagena Protocol on Biosafety of the CBD, providing detailed information on inserted genes of GM plants that are marketed globally. GMOs are also specified by unique identifiers, international codes developed by the OECD [19] that are specific to each GMO brought to market in accordance with Commission Regulation (EC) No. 65/2004 [20].

Ideally, similar to the detection of classic GM plants, a method for detection of a genome-edited plant will also allow its required identification, which includes the allocation to a specific developer. The unambiguous assignment of a detection result to a specific product, and thus developer, is sufficient for identification. A recently conducted legal assessment supports this conclusion [21]. Either the necessary information for such an assignment is provided directly by the developer, or a connection can be established between the sequence alterations and a marketed product by alternate ways, e.g., by consulting regulatory databases or information published in patents. If adequate sequence information is made available, distinct, unique genomic changes can be analytically detected [22]. These changes may either be (several) intended mutations or a combination of intended mutations with unintended changes that are simultaneously present in a genome-edited plant [23,24,25]. Plant genes with a low genetic distance may not be separable, a phenomenon frequently observed in classical breeding [26]. Similarly, mutations are expected to be genetically linked if located in the immediate vicinity. Moreover, genome editing has the potential to introduce several specific modifications in a genome. In either case, these combinations of mutations can be analytically detected. The purely analytical detection and identification of a genome-edited plant or product still relies on the uniqueness of the specific sequence alteration(s). Different sources of information might be consulted to establish knowledge about a unique combination of genetic modifications that is sufficient for identification. This complementary information can also be useful to establish whether genome editing was applied in the breeding process.

## 3. Data Requirements and the International Regulatory Landscape

Whenever a GMO product is notified for regulatory approval in the EU, the developer has to submit a dossier that, inter alia, needs to contain sequence information and a proposal on a suitable method for its unambiguous detection, identification and quantification. The developer also has to supply sample material. Only after a thorough validation process, the method is published in the EU Database of Reference Methods for GMO Analysis (GMOMETHODS) [27]. It is then available to be applied to enforce traceability and other regulatory requirements such as mandatory GMO labeling; besides, reference material must be readily available.

For GMO and genome-edited products not yet submitted for regulatory approval in the EU, the abovementioned obligations do not apply. However, GMOs may have already been authorized for environmental release and commercial marketing in one or more countries other than the EU. Thus, such products may potentially be present, e.g., at low levels, in imported seeds or agricultural commodities [28]. Similarly, genome-edited products might be on the global market, with our without authorization, and are defined as GMOs following the decision of the European Court of Justice [4]. The presence of unauthorized GMOs on the EU market may lead to trade disruptions and cause significant financial damage due to the rejection or the destruction of goods [29].

Unauthorized products cause significant challenges to European authorities, as only an authorization request ensures immediate access to the regulatory information required for traceability and labeling. Situations of asynchronous authorization present challenges for detection [30], as no official information may be accessible to European regulators due to differences in national biosafety laws and the respective authorization procedures and pipelines [8,9]. The specifics of regulatory frameworks for biosafety operating in a number of non-EU countries may foresee that certain GMOs and, in particular, a wide range of genome-edited products will not require regulatory approval for market release [8]. Among others, some North and South American countries, including the USA, Argentina, Brazil, Chile, Colombia and Paraguay, as well as Australia do not regulate certain types of genome-edited plant products [9]. In particular, this applies to those products which do not contain transgenic DNA or from which the transgenic DNA has been removed (applicable in Argentina and other South American countries; [31]) or that contain genetic modifications that may also occur by spontaneous mutations (applicable for plants in the USA [32]). Canada has a product-oriented biosafety regulation based on novelty as a trigger and regulates all plants with novel traits irrespective of the method of their development. The Canadian regulatory framework also covers genome-edited plants if they display novel traits, but without requiring an identification method for approval [8].

## 4. Available Sources of Data for Genome-Edited Plants

Identification of a genome-edited plant or food/feed product is facilitated by the availability of sufficient information that allows determination of its precise origin and identity. The most relevant information obviously includes the genetic modification process and, thereby, the resulting genomic and phenotypic attributes of a genome-edited plant. However, this information is owned by the developer. As a consequence, for many genome-edited plants that reach market status outside the EU, this information is difficult to retrieve and, consequently, potentially not available to competent national authorities and their laboratories. Moreover, genome editing has only been used in recent years to modify plants for food and feed use, and these plants are currently just about to enter the global market. In contrast to well-characterized GM plants with known traits that have been used commercially for some time, less knowledge and information are typically available for genome-edited plants. Despite these constraints, open-access databases are available that provide relevant information on genome-edited plants that can be used for the development of detection methods.

Databases of non-EU authorities (e.g. U.S. Department of Agriculture - Animal and Plant Health Inspection Service—USDA-APHIS, Health Canada Novel Food Information, and Australian Office of the Gene Technology Regulator) or general information about genome-edited plants and the developers can be found in application documents that are made publicly available to some extent. However, these information sources are limited regarding sequence information and detection methods. Significant bottlenecks are that sequence information is usually not provided in publicly accessible databases, and that specific regulatory regimes in non-EU countries frequently do not foresee the submission of detection methods. Thus, even upon availability of suitable sequence information, development and validation of methods require additional efforts and resources.

The BCH serves as a global platform to exchange information on living modified organisms (LMO) according to the Cartagena Protocol on Bio-safety of the CBD [33]. The parties to the protocol are obliged to share, via the BCH, information pertinent to regulated products. In addition, countries that are not parties to the protocol, such as the USA, Canada and Australia, use the BCH to exchange and share information on the GMO products developed and regulated in these countries. The BCH is thus seen as a tool that is contributing to “governance by transparency” [34]. The BCH database [18] can be used as a data source for information on GMOs, including genetic elements, traits, authorization status or field trial status. It generally provides information on the role and function of mutated genes but no sequence data. The BCH database already contains information about some genome-edited plants, e.g., herbicide-tolerant canola or maize with tolerance to abiotic stress (Record IDs 110268 and 115124). Cross-reference to detection methods is provided, if available.

EUginius [35] is a European initiative for a unified database system and a valuable data source with regard to both classic GMOs and genome-edited plants. The database went online 2014 and is hosted by the German Federal Office of Consumer Protection and Food Safety and the Wageningen Food Safety Research of Wageningen University. It presents not only details on GMOs but, if available, also detection method protocols, usually derived from the GMOMETHODS database [36]. EUginius allows a full text search and offers advanced search features. For genome-edited plants, the database contains taxon-specific detection methods to identify the modified plant species but does not include detection of the specific modification. However, if freely available, for instance in the scientific literature, EUginius gives access to information on the mutation caused by the genome editing technique. Since EUginius focuses on information and issues regarding the detection of GMOs, it gives a good example and model for a state-of-the-art database that could, in the future, also provide useful data regarding detection of genome-edited plants.

Patent applications may also provide information about genome-edited plants, enabling the development of detection methods, as they are relevant data sources containing information on the genomic and phenotypic features of new genome-edited plants. Espacenet [37], which is freely accessible to the public, and European Patent Office QUEry (EPOQUE), a search software developed by the European Patent Office for use by patent examiners integrating all patent databases accessible to the European Patent Office, are the two most relevant platforms for patent data in the EU. Both patent databases allow access to almost the same datasets of information and enable searches for any patent worldwide. With regard to genome-edited plants, the most important technical data presented in patents are: a description of the new invention, application range, examples of (genome-edited) plants/products, vectors used for transformation, modified genomic sequence data, target genes and a description of the expected mutations, a graphical display of the new invention and reference to the original publication(s). Not all patent documents contain the same type of technical data; based on their availability, especially the genomic sequence data and the description of the introduced mutations could support the development of detection methods. Critical examination of patents covering products with market potential reveals an inhomogeneous picture with regard to the presented technical data. Not all patents present genomic sequence data as well as information about the specific mutations introduced by the invention.

Another valuable information source is the scientific literature, which may provide useful details regarding genome-edited plants that might enter the market in the future. It will be important to systematically collect scientific publications in a comprehensive database and to link them to other available information in a systematic way. Other useful components of an appropriate system of information exchange are databases like the Plant Genome Editing Database (PGED) [38] hosted by the United States Boyce Thompson Institute that gathers CRISPR)/Cas research data, or the CrisprGE database [39] located at the Bioinformatics Centre, Institute of Microbial Technology in Chandigarh, India. Both databases collect information about target genes and genomic modification of many different plant species that could be useful for detection purposes. Product information on newly developed genome-edited plants can further be found in developer specifications published online, via newsletters or other media channels. Sometimes such information contains statements concerning the market maturity of certain products or on the general product portfolio.

## 5. Using Available Data Sources: Practical Considerations

Two genome-edited products, a herbicide-tolerant canola and a powdery mildew-resistant wheat, were chosen due to their commercialization status and market relevance to illustrate the benefits and drawbacks of the available sources of information. These examples also highlight that different aspects of the associated difficulties need to be tackled as regards detection and, specifically, identification.

Recently, a method has been developed to detect a specific mutation in SU (sulfonylurea and imidazolinone herbicide-tolerant) canola varieties [22]. The authors used information available in the public domain, namely EUginius, BCH and Health Canada Novel Food Information. In their publication, Chhalliyil et al. [22] described sequence information and the introduced mutations for the tested canola varieties. This description of the genome-edited plant has allowed the development of a robust, validated detection method. The limitations of the method were discussed by the European Network of GMO Laboratories (ENGL; [40]), who pointed out that the possibility of developing a detection method depends on the availability of information. In this case, sufficient information could be collected to devise a detection method. ENGL also stated that the method is highly specific due to the combined detection of two mutations [40]. However, as stated by ENGL, the most important limitation relates to the possibility of unequivocally identifying that the product is a result of a genome editing approach. This limitation stated by ENGL can be overcome if an analytical result can be directly linked to a specific product and developer. This is the case when adequate information is available; for instance, if the method for introducing the genetic modification was disclosed by the developer. A similar conclusion is reflected by a legal analysis [21] concerning the detection method published by Chhalliyil et al. [22] for herbicide-tolerant canola.

Another genome-edited plant with marketing potential is bread wheat that shows resistance to powdery mildew due to a transcription activator-like effector nuclease (TALEN)-induced mutation of three *Mildew Locus O* (*MLO*) gene homoeoalleles. The applied genome editing technique, including details of the genetic modification (gene knockout) of *MLO* genes of the bread wheat variety, is described by Wang et al. [41]. There is no detection method available as for the herbicide-tolerant canola, but data on the modified sequence can be found in Wang et al. [34]. In addition, an international patent application (WO 2017/013409 A2) was filed by the Chinese Academy of Sciences [42]. Sufficient sequence information is provided to develop a detection method for the modified *MLO* genes contained in the wheat variety, as described in the patent. If the particular genome-edited bread wheat entered the market and reference material (e.g., food/feed samples, seeds, plant material) was available, the exact sequence of the mutations in all three *MLO* gene homoeoalleles could be detected by a PCR-based method, and a connection to the marketed product could be established. The assistance of competent authorities or the developer’s cooperation might be sought to gain access to relevant information and reference material, a situation comparable to the procedure when non-authorized transgenic GMO have inadvertently entered the market.

## 6. Discussion

The availability of sequence information is the main requirement to devise and implement analytical detection by EU institutions. Online databases, patent applications and scientific literature may offer details on the specific genetic modification(s) in genome-edited plants. The information collected in these sources is helpful for those genome-edited plants that have not yet been subject to any authorization procedure or for which no detection method is available. However, the available data may be of varying completeness, specificity and reliability. Ideally, the information will be sufficient to develop an unequivocal detection method. This would typically be the case when a notification for authorization of a genome-edited product is submitted to EU authorities. In this case, a detection method according to the EU standards is submitted and appropriate reference materials are available; once authorization is granted, they are accessible to reference laboratories. In another case, adequate information to devise a detection method is available when genome-edited products are not (yet) notified for use in the EU but regulated or authorized by other countries, particularly by countries that mandate GM labeling requirements. Finally, if no validated detection method exists, the available information may be fully sufficient to devise such a method, as has recently been published by Chhalliyil et al. [22]. Relevant information is either available publicly from different sources or may be shared in the framework of existing information exchange systems, such as the BCH or other official channels for international cooperation (e.g., OECD working groups). In addition, the respective products may already be used commercially, and adequate material may be obtained for use as reference, provided that the developer does not object to such a use. The availability of material that is suitable as a reference can only be determined on a case-by-case basis. It is recommended that reference material be obtained directly from the developer whenever possible.

The recent publication by Chhalliyil et al. [22] illustrated that different types of information can be used to develop a detection method for a specific product and that the publicly available information may be sufficient to design appropriate detection methods. The example also showed that despite high specificity, identification is nevertheless challenging. While unequivocal detection is the basis for identification, the latter might not be strictly a matter of detecting a certain sequence change.

Any information that is found by consulting the available data sources may be used to increase the certainty of successfully identifying a genome-edited plant. Like in a jigsaw puzzle, every piece of information adds to the evidence that indeed genome editing was involved in the development of a specific product. The more comprehensive the knowledge about a new genome-edited plant, the higher the probability of identification. Data relevant for identification can be collected from diverse databases, patent applications, scientific literature or information published by the developer. These valuable data sources are usually freely available. The respective information is useful in the case when genomic and phenotypic data are not made available by the developer.

The most challenging case is when some information on a new product is available but has to be collated from different information sources. For instance, only general information might be available from non-EU authorities concerning a new product. Examples are submissions to determine the regulatory status of a new product, such as information published by the USDA-APHIS in the framework of submissions according to the “Am I regulated?” (AIR) process [32], or similar information released by South American countries operating a decision-making process (e.g., [31]). Such information may then be supplemented by actively consulting other sources, such as patent applications, information from the scientific literature or other public sources including information published by the developer. These sources may include the necessary (sequence) information to devise a detection method. A variety of databases can be a valuable source for relevant information to help identify new genome-edited plant varieties. To be able to use the available data in an efficient way, information from different sources should be collected in a standardized manner. For efficient use, cross-references in a frequently updated database are helpful to support quick access to complete information. Resources need to be allocated to host, maintain and update such a database. It is important that data are easily accessible, highly reliable and regularly checked for completeness. Frequent updates can further guarantee that the information provided by databases remains state-of-the-art. A useful example for a database that focuses on datasets for detection and identification purposes is the EUginius database [36].

## 7. Outlook

The approaches used currently for detection and identification of classic GMOs and GM food and feed products may guide the development of similar approaches for genome-edited products. However, concerning the identification of genome-edited plants, specific considerations are necessary. The availability of information concerning the introduced genetic modification(s) is a required starting point to develop detection methods, and, moreover, to establish the connection between an analytical detection result and a specific product. In addition, unique and product-specific modifications allow establishment of the identity of a genome-edited product.

The detectable presence of characteristic modifications may indicate the identity of a genome-edited product with a sufficiently high level of probability. For instance, if it is known that genome-edited powdery mildew-resistant bread wheat is on the global market, the simultaneous detection of the characteristic known mutations in the three *MLO* gene homoeoalleles in a wheat sample would be a strong indication that this sample contains genome-edited wheat.

To unequivocally establish a direct link between a detected sequence and a specific product, diverse sources of information may feed into a process to identify a genome-edited product. This approach would highly depend on the cooperation of competent authorities, governmental agencies, researchers and companies. The key element of such a network is the voluntary disclosure of information by the developers and authorities involved in pre-submission consultations; publicly available data, e.g., from the scientific literature and patents, are also valuable information sources. It is advisable to establish a mechanism for checking the quality of the collected information.

According to Jordan et al. [43], a cooperative governance network including developers and authorities could support the information exchange required to facilitate the international trade of genome-edited products. A number of such cooperative networks have been successfully established for the governance of food products, e.g., coffee or fish products. They have performed well in comparable situations by defining broadly acceptable criteria for products and processes, and establishing institutional capacity when government-based regulations are neither established nor sufficient [43].

An anticipatory framework may act as a trusted source that provides for a platform of information exchange with defined mechanisms for submission and exchange, quality checks and provisions for the disclosure of information. The suggested framework should take advantage of various relevant best practice examples: stewardship programs [12], cooperative governance networks [43], information exchange [34], information disclosure practices (patents, USDA-APHIS AIR) and appropriate database design [36]. International cooperation and support is required based on an adequate structure. For this, the necessary resources need to be made available to ensure long-term operability.

The use of this detection and identification framework depends on the regulatory landscape. Concretely, if products of genome-editing are regulated, the respective statutory authorities need to decide on the level of information that is required for identification. Detection methods and criteria defined by the authorities also allow the implementation of voluntary labeling schemes. Development and validation of appropriate detection and identification methods should be undertaken by developers (for products that are notified) and by governmental institutions that are provided with the necessary resources (for non-authorized products or when regulations do not require an analytical method).

Given the rapid advances in development, fast implementation of an international anticipatory detection and identification framework for collaboration and the collation of voluntarily disclosed information is advisable. By exploiting a variety of different information sources in a systematic manner, and by establishing which level of information is sufficient to conclude on the identity of a particular genome-edited product, it is reasonable to expect that identification of genome-edited food or feed products is possible.

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
