# Peer review of "Genome-Edited Plants: Opportunities and Challenges for an Anticipatory Detection and Identification Framework"

_foods, 2021, doi:10.3390/foods10020430_

Round 1

Reviewer 1 Report

Generell comments

There is an ongoing discussion on legal and scientific requirements for the detection and identification of genome edited plants. Specifically, plants developed by new molecular techniques like CRIPSR/cas resulting in site-directed nuclease type 1 or 2 (SDN1/2) alteration in the target genome are challenging for both regulatory authorisation and – if required - law enforcement. The manuscript contributes to the debate in a more general way by presenting some cornerstones for a “anticipatory detection and identification framework for genome-edited plants based on broad information gathering and exchange”. The contribution provides a shallow treatment of a complex situation. The basic proposal is the establishment of a database collecting all information for genome-edited plants „easily accessible, highly reliable and regularly checked“ (line 301). There is no information given for resources and support required. One existing database (EUginius) is mentioned as a positive example of such a kind of database (line 304), so why is another database necessary? Finally, the authors propose the development of criteria “that are deemed sufficient to unambiguously conclude on the identity of a particular genome-edited product.” Without providing details on such criteria in the manuscript, the authors conclude that such criteria can be the starting point for an “anticipatory detection and identification framework“ (line 327-329). As vague as the framework is, as little convincing the opportunities and challenges remain. It sounds like “Collect everything and see what you can make out of it” at present stage. Overall, the manuscript is not a scientific or law enforcement breakthrough, but could be nevertheless an interesting contribution in the ongoing discussion.

Detailed comments:

20-21: If no information (and no detection method) is made available, such plants cannot even be approved. The labelling and traceability mentioned before, which this sentence relates to, are designated to GMOs authorized for the EU marked. Unauthorized GMO need not be labelled, as they are prohibited from being used in general.

27-29: What’s the benefit for countries who do not require such enforcement?

33: A definition of genome editing explaining SDN type 1,2, and 3 would be helpful. This relates to existing difficulties in distinguishing SDN1/2 alterations from other sources of mutations.

36-37: Proposal for improvement:  …an analytical method able to unambiguously demonstrate their identity.

39: The legal EU requirements use the term ‘plan’ instead of ‘measures’ for PMEM.

42ff: IP control systems have a different legal status comparted to authorization procedures if they are voluntary like in the GMO-free value chain. I propose at least starting here with a new paragraph, and please mention the difference.

44: The verb ‘support’ instead of ‘ensure’ might be a better descriptor.

61-63: What is the difference to unintended mutations occurring during classical breeding?

75-77: There are countries under the CBD umbrella that do not regulate genome edited plants. Do I miss the point here? See also lines 134 ff.

100-103: Proposal for improvement: „Hence identification may be defined as the unambiguous allocation to a specific plant or product (event-specific detection) and a specific developer. Unless there is a unique combination of sequence alterations in a genome-edited plant this will not directly be achieved via a detection method. but might be possible by utilizing additional information” While the purely analytical identification of a genome edited plant or product seems challenging, if not impossible, the allocation to a specific plant or developer might be possible by utilizing additional information.

106-107: Proposal for improvement: „If adequate sequence information is made available to regulatory bodies, distinct, unique genomic changes can be detected serve to identify a particular genome-edited plant or derived product with an acceptable level of certainty”

112: The sentence should be deleted or changed to: “A genome-edited product can be identified detected based on the detection of unique sequences [18]”. This is in line with my next remark:

113-114: A question to “The unambiguous assignment of a detection result to a specific product, and thus developer, is sufficient for identification. A recent legal assessment supports this conclusion”: When can there be an unambiguous assignment of a detection result to a specific product? Especially the case mentioned in reference 18 is currently highly disputed. In lines 275-277 it is stated that "While unequivocal detection is the basis for identification, the latter might not be strictly a matter of detecting a certain sequence change."

116-120: Is such a regulatory approval step possible for SDN1/2 genome edited plants in the EU?

127: The OECD reference [21] mentions only transgenic and not genome edited plants.

140-141: Proposal for improvement: “There are also countries such as Canada that regulate genome-edited products for biosafety if they display novel traits, but without requiring an identification method for approval [23].” This trait-based approach to regulation in Canada is not specific for genome-edited products, but applies to all new varieties.

143: The EFSA reference [27] addresses basically transgenic and not genome edited plants, which are not intended to be placed on the EU market, and where information from other legal authorization requirements might be available. This could be e.g. ‘Golden Rice’. EFSA's task is safety assessment and not GMO trade, detection, and identification.

145-146: Proposal for improvement: “Identification Detection of a genome-edited plant or food/feed product is facilitated by sufficient information that allows determining its precise origin and identity.” I would rather see detection than identification. In my opinion, the origin cannot be identified analytically if the origin/cause of the mutation is meant here.

152-153: I do not understand the conclusion: “Moreover, genome editing has only been employed in recent years to modify plants for food and feed use. Therefore, less information is typically available for such plants as compared to classic GMOs.” For each individual GMO/genome-edited variety, the information available is independent of how long the technology has been used.

168-169: Please mention the BCH record numbers for the examples. “It already contains information about some genome-edited plants, e.g. herbicide-tolerant canola or maize with tolerance to abiotic stress.”

217-218: Proposal for improvement since reference 18 is disputed: “Recently, a method has been developed to detect the specific mutation in SU (sulfonylurea and imidazolinone herbicide-tolerant) canola varieties [18], declared by the authors to be developed via oligonucleotide directed mutagenesis (ODM). As far as I know the developer declared the opposite. The referenced paper may have shown how to develop a detection method. The problem of identification in this specific case still exists though.  

226-231: Allthough the ENGL statement is correct: “However, as stated by ENGL the most important limitation relates to the possibility to unequivocally identify that the product is a result of a genome editing approach.”, it should be better separated from the next “However…” sentence by the authors (and which might not be shared by ENGL).

240-246: I find the conclusion unconvincing: There is no information on the sequence, but according to the authors this could be investigated - and a detection method be developed - when the wheat comes onto the market... But either the wheat is officially approved, in which case the sequence, method, reference material etc. would have to be made available anyway, or it reaches the EU illegally - but then authorities can only speculate when and where to look for such wheat. Is this a realistic example – e.g. for trade data or willingness of developers and Chinese authorities to cooperate with EU authorities?

272: Proposal for improvement since reference 18 is disputed: “The recent publication by Chhalliyil et al. [18] demonstrates indicated….”

307-309: I highly question the conclusion and recommend at least a deletion: “In most cases, the The detectable presence of characteristic modifications may indicate the identity of a genome-edited product with a sufficiently high level of probability.” As elaborated clearly in other parts of the manuscript, a detection of a known mutation does not necessarily indicate the identity a genome-edited product with a “sufficiently high level of probability”. It is also highly debatable, what a “sufficiently high level of probability” shall be. Either, a product is unambiguously identified, or not. This contradiction is also supported in line “The most important seems to be to define thresholds in relation to the respective probabilities that are deemed sufficient to accept a conclusion on the identity of a specific genome-edited product. “ The manuscript would benefit from quantitative threshold examples.

Author Response

R…Reviewer’s comments

A…Authors’ reply

R: Generell comments

R: There is an ongoing discussion on legal and scientific requirements for the detection and identification of genome edited plants. Specifically, plants developed by new molecular techniques like CRIPSR/cas resulting in site-directed nuclease type 1 or 2 (SDN1/2) alteration in the target genome are challenging for both regulatory authorisation and – if required - law enforcement. The manuscript contributes to the debate in a more general way by presenting some cornerstones for a “anticipatory detection and identification framework for genome-edited plants based on broad information gathering and exchange”. The contribution provides a shallow treatment of a complex situation. The basic proposal is the establishment of a database collecting all information for genome-edited plants „easily accessible, highly reliable and regularly checked“ (line 301). There is no information given for resources and support required. One existing database (EUginius) is mentioned as a positive example of such a kind of database (line 304), so why is another database necessary? Finally, the authors propose the development of criteria “that are deemed sufficient to unambiguously conclude on the identity of a particular genome-edited product.” Without providing details on such criteria in the manuscript, the authors conclude that such criteria can be the starting point for an “anticipatory detection and identification framework“ (line 327-329). As vague as the framework is, as little convincing the opportunities and challenges remain. It sounds like “Collect everything and see what you can make out of it” at present stage. Overall, the manuscript is not a scientific or law enforcement breakthrough, but could be nevertheless an interesting contribution in the ongoing discussion.

A: We thank the reviewer for pointing out some parts of the manuscript that needed further clarification. In particular, we have sought to refine and expand on the opportunities and challenges.

Concerning the support required, we believe that at this stage it is most important to strive for international cooperation, and thus support by an international body would be highly welcome.

Indeed, as an example EUginius was chosen, as to date it is to our knowledge the most comprehensive database that includes not only information on plants already approved or with a pending approval on a broad basis but also provides access to information that is particularly relevant for detection and identification, e.g. from the scientific literature or from patents. Where available, it furthermore gives detailed information on the modification. We thus agree with the reviewer’s remark that the proposed database to support the framework may build upon EUginius. Given the indispensable international dimension it will, however, be necessary to rethink the organisational structure of operation. In this context, it is also important to consider that the availability of the resources necessary for long-term operation needs to be ensured.

Thank you for your comment concerning the criteria. We have rephrased the “Outlook” part of the manuscript and have specified the “starting point”. We hope that the revisions have made the message clearer, and that it is also more transparent why a particular kind of information should be collected and how the collected information could be used.

R: Detailed comments:

R: 20-21: If no information (and no detection method) is made available, such plants cannot even be approved. The labelling and traceability mentioned before, which this sentence relates to, are designated to GMOs authorized for the EU marked. Unauthorized GMO need not be labelled, as they are prohibited from being used in general.

A: Thank you for indicating that more precision is needed. We have rewritten the abstract, the sentence has been removed.

R: 27-29: What’s the benefit for countries who do not require such enforcement?

A: The rewritten abstract includes a sentence concerning the global benefit, which mainly concerns to avoid trade disruptions and to facilitate labelling schemes, also beyond enforcement (e.g. identity preservation schmes). We have also elaborated on this in the Introduction and in Chapter 3.

R: 33: A definition of genome editing explaining SDN type 1,2, and 3 would be helpful. This relates to existing difficulties in distinguishing SDN1/2 alterations from other sources of mutations.

A:We have added a short explanation on the different types of alterations caused by genome editing. We have also included a reference to a comprehensive report on “New Techniques” for interested readers as background information.

R: 36-37: Proposal for improvement:  …an analytical method able to unambiguously demonstrate their identity.

A: Thank you for your proposal. We have changed the text to fit the corresponding European Commission Implementing Regulation.

R: 39: The legal EU requirements use the term ‘plan’ instead of ‘measures’ for PMEM.

A: We have changed the term accordingly.

R: 42ff: IP control systems have a different legal status comparted to authorization procedures if they are voluntary like in the GMO-free value chain. I propose at least starting here with a new paragraph, and please mention the difference.

A: Thank you for this indication. It is indeed an important difference, and we have therefore taken into account your proposal.

R: 44: The verb ‘support’ instead of ‘ensure’ might be a better descriptor.

A: The term has been changed in the revised text.

R: 61-63: What is the difference to unintended mutations occurring during classical breeding?

A: Thank you for bringing up this point. We agree that this aspect is generally relevant in the genome editing discussion but these differences are not the topic under debate in our manuscript. It is nonetheless important that any alteration in the genome might be harnessed for detection. The text has been expanded accordingly here and later on in the manuscript.

R: 75-77: There are countries under the CBD umbrella that do not regulate genome edited plants. Do I miss the point here? See also lines 134 ff.

A: Thank you for this very valid point. There is an ongoing debate on this topic, as members of the CBD would in principle be obliged to regulate. In our view, elaborating on this controversial discussion in the manuscript is beyond its scope. Your comment inspired us, however, to elaborate on the BCH later in the text.

R: 100-103: Proposal for improvement: „Hence identification may be defined as the unambiguous allocation to a specific plant or product (event-specific detection) and a specific developer. Unless there is a unique combination of sequence alterations in a genome-edited plant this will not directly be achieved via a detection method. but might be possible by utilizing additional information” While the purely analytical identification of a genome edited plant or product seems challenging, if not impossible, the allocation to a specific plant or developer might be possible by utilizing additional information.

A: We have elaborated on the issue of identification, and have paid special attention to the used terms to avoid potential misconceptions. Please refer to the revised text in the manuscript.

R: 106-107: Proposal for improvement: „If adequate sequence information is made available to regulatory bodies, distinct, unique genomic changes can be detected serve to identify a particular genome-edited plant or derived product with an acceptable level of certainty”

R: 112: The sentence should be deleted or changed to: “A genome-edited product can be identified detected based on the detection of unique sequences [18]”. This is in line with my next remark:

A: Please see the above considerations, and please refer to the revised text.

R: 113-114: A question to “The unambiguous assignment of a detection result to a specific product, and thus developer, is sufficient for identification. A recent legal assessment supports this conclusion”: When can there be an unambiguous assignment of a detection result to a specific product? Especially the case mentioned in reference 18 is currently highly disputed. In lines 275-277 it is stated that "While unequivocal detection is the basis for identification, the latter might not be strictly a matter of detecting a certain sequence change."

A: Thank you for pointing out this need for further clarification. We have responded by revising the text, taking into consideration the remarks. We hope that by this we have explained better that there might be cases where the analytical detection also leads to identification. In contrast to classic GMOs, for genome-edited plants it is likely that neither detection nor identification can be achieved without prior information. If, however, based on this information it is possible to devise an unambiguous detection method based on specific alterations, the product can be assigned to a specific developer.

R: 116-120: Is such a regulatory approval step possible for SDN1/2 genome edited plants in the EU?

According to the current regulations, the applicant has to comply with the steps of regulatory approval, including a suitable method for detection. Taking into consideration the above-mentioned, we assume that in most, if not all, cases it will be possible to design a suitable method. For this, it will be necessary to look at the genomic constitution of the plant beyond a sole SNV.

R: 127: The OECD reference [21] mentions only transgenic and not genome edited plants.

A: Yes, we agree that we need to be more precise. We have removed the reference to genome-edited plants in the indicated sentence and have added a more precise explanation below.

R: 140-141: Proposal for improvement: “There are also countries such as Canada that regulate genome-edited products for biosafety if they display novel traits, but without requiring an identification method for approval [23].” This trait-based approach to regulation in Canada is not specific for genome-edited products, but applies to all new varieties.

A: The explanation on the Canadian regulation practice has been rephrased.

R: 143: The EFSA reference [27] addresses basically transgenic and not genome edited plants, which are not intended to be placed on the EU market, and where information from other legal authorization requirements might be available. This could be e.g. ‘Golden Rice’. EFSA's task is safety assessment and not GMO trade, detection, and identification.

A: Following the remark, we have removed the sentence.

R: 145-146: Proposal for improvement: “Identification Detection of a genome-edited plant or food/feed product is facilitated by sufficient information that allows determining its precise origin and identity.” I would rather see detection than identification. In my opinion, the origin cannot be identified analytically if the origin/cause of the mutation is meant here.

A: Our rationale does not include the identification of the origin/cause of the mutation by analytical means but by using information to connect an analytical result with a specific product and developer. Based on the available information, it can furthermore be established that genome editing was used in the breeding process. Either this information is directly provided by the developer, or it is established by experts based on agreed criteria and on collated pieces of information. For this reason, we have not changed the term according to the proposal.

R: 152-153: I do not understand the conclusion: “Moreover, genome editing has only been employed in recent years to modify plants for food and feed use. Therefore, less information is typically available for such plants as compared to classic GMOs.” For each individual GMO/genome-edited variety, the information available is independent of how long the technology has been used.

A: Here, we have added more information and hope to have clarified the message.

R: 168-169: Please mention the BCH record numbers for the examples. “It already contains information about some genome-edited plants, e.g. herbicide-tolerant canola or maize with tolerance to abiotic stress.”

A: We have added selected BCH record numbers.

217-218: Proposal for improvement since reference 18 is disputed: “Recently, a method has been developed to detect the specific mutation in SU (sulfonylurea and imidazolinone herbicide-tolerant) canola varieties [18], declared by the authors to be developed via oligonucleotide directed mutagenesis (ODM). As far as I know the developer declared the opposite. The referenced paper may have shown how to develop a detection method. The problem of identification in this specific case still exists though.

A: We have removed the second part of the sentence, as this aspect is not our point of discussion.

R: 226-231: Allthough the ENGL statement is correct: “However, as stated by ENGL the most important limitation relates to the possibility to unequivocally identify that the product is a result of a genome editing approach.”, it should be better separated from the next “However…” sentence by the authors (and which might not be shared by ENGL).

A: The text has been rephrased accordingly.

240-246: I find the conclusion unconvincing: There is no information on the sequence, but according to the authors this could be investigated - and a detection method be developed - when the wheat comes onto the market... But either the wheat is officially approved, in which case the sequence, method, reference material etc. would have to be made available anyway, or it reaches the EU illegally - but then authorities can only speculate when and where to look for such wheat. Is this a realistic example – e.g. for trade data or willingness of developers and Chinese authorities to cooperate with EU authorities?

A: Thank you for your comment; the text was not clear enough here. Please refer to the rephrased paragraph. Importantly, the cited publication (Wang et al [34]) contains sufficient sequence information to develop a detection method. To add a best practice example, the revised text includes that EU authorities have in the past collaborated with developers and other authorities in order to obtain information about events that have entered the supply chain inadvertently.

R: 272: Proposal for improvement since reference 18 is disputed: “The recent publication by Chhalliyil et al. [18] demonstrates indicated….”

A: The dispute mentioned relates to the method used and not the detection of a specific sequence change, as laid out in the text. That the test is able to detect a specific sequence change in a particular product has been confirmed by ENGL. We refrain from changing the text.

R: 307-309: I highly question the conclusion and recommend at least a deletion: “In most cases, the The detectable presence of characteristic modifications may indicate the identity of a genome-edited product with a sufficiently high level of probability.” As elaborated clearly in other parts of the manuscript, a detection of a known mutation does not necessarily indicate the identity a genome-edited product with a “sufficiently high level of probability”. It is also highly debatable, what a “sufficiently high level of probability” shall be. Either, a product is unambiguously identified, or not. This contradiction is also supported in line “The most important seems to be to define thresholds in relation to the respective probabilities that are deemed sufficient to accept a conclusion on the identity of a specific genome-edited product. “ The manuscript would benefit from quantitative threshold examples.

A: Taking up the reviewer’s comments, we have substantially changed the Outlook section of the manuscript. We hope that by this our conclusions on identification become clearer and more transparent. We have furthermore given an example where a “sufficiently high level of probability is given” and decided to omit the term “thresholds” because we felt that it was inappropriate to explain our conclusions.

Reviewer 2 Report

Report for foods

Manuscript ID Foods-1041972

Title ‘Genome-edited plants: Opportunities and challenges for an anticipatory detection and identification framework”

General comments: -

In general, the topic is very interesting and has a great impact. I found the paper to be overall well prepared. The manuscript is well organized, but I have some minor comments.

Detailed comments: -

Keywords:

Please take off the word traceability                                               

Abstract

This section is missing the direct results from the current study. Please state the important findings from this study.

Line 22 Please avoid using the personal pronouns

             Please change we propose to to… This study proposed to

Apply this rule through the manuscript

Introduction

This section needs to be enriched and expanded by adding more background about the topic

Line 78 : 2- Detection may not fully ensure identification … This is not clear

Line 142 please change result in challenges to result in many challenges

  1. Using availability data sources: Practical considerations

Line 245-246: In this context, the assistance….be sought. This part is confusing, please reword it.

Line 247-249 It is important ….. products. Please take off this part, (it is common knowledge and doesn’t add a piece of new information).

  1. Discussion

Line 270:please change we recommend that to ….It is recommended that

7- outlook

Line 306: Please change in systematic manner to in a systematic manner

Author Response

R…Reviewer’s comments

A…Authors’ reply

General comments:

In general, the topic is very interesting and has a great impact. I found the paper to be overall well prepared. The manuscript is well organized, but I have some minor comments.

Detailed comments:

Keywords:

R: Please take off the word traceability

A: The word traceability was deleted from the list of keywords.

Abstract

R: This section is missing the direct results from the current study. Please state the important findings from this study.

A: Thank you. We have rewritten the Abstract taking into consideration this valuable comment for improvement.

R: Line 22 Please avoid using the personal pronouns

Please change we propose to to… This study proposed to

Apply this rule through the manuscript

A: The phrase was changed accordingly. The manuscript was checked for the use of personal pronouns, and the text was adapted where necessary.

Introduction

R: This section needs to be enriched and expanded by adding more background about the topic

A: We have added more details on genome editing, and have also elaborated on enforcement vs. voluntary labelling schemes.

R: Line 78 : 2- Detection may not fully ensure identification … This is not clear

A: The title of this Chapter was amended and aligned to the other headings in the manuscript.

R: Line 142 please change result in challenges to result in many challenges

A: The phrase has been deleted as we found it inappropriate at that point in the text.

Using availability data sources: Practical considerations

R: Line 245-246: In this context, the assistance….be sought. This part is confusing, please reword it.

A: The sentence was expanded for more clarity.

R: Line 247-249 It is important ….. products. Please take off this part, (it is common knowledge and doesn’t add a piece of new information).

A: We have deleted this part following the reviewer’s recommendations.

Discussion

R: Line 270:please change we recommend that to ….It is recommended that

A: The text was changed accordingly.

Outlook

R: Line 306: Please change in systematic manner to in a systematic manner

A: Please refer to the rewritten “Outlook” section. The sentence in question has been deleted in the revised version of the manuscript.

Round 2

Reviewer 1 Report

The manuscript has substanially improved. I appreciate that the authors considered most of my recommendations. However two important issues remain:

First, I strongly recommend deleting both citation 20 and 21. This step would not not alter the scientific contend of the manuscript.

Reasoning:

1a) Line 170ff: is referring to the following sentence in Berteau 2019 ““Recurrent detection of differences (on target and off-targets) and close PAM16,17 sequence [s] would form a strong predictor, not to say an unambiguous signature of the employed technique.” The ‘comprehensive’ review of Berteau 2019 (as a non-peer review book chapter) is disputed, e.g. by the German Biosafety Commission ZKBS 2020: “Bertheau suggests that unintentional and spontaneous mutations could be distinguished from intended alterations and claims that the applied New Breeding Technique could be identified as well, all by characterization of the organism, especially through phenotype characterization and different sequencing procedures. This suggestion lacks a scientifically sound explanation and is unfeasible according to the state-of-the-art of science and technology.” For more details: https://www.zkbs-online.de/ZKBS/EN/01_Aktuelles/ZKBS-commentary%20on%20Y.%20Bertheau%20(2019)/ZKBS-commentary%20on%20Y.%20Bertheau%20(2019)_node.html;

1). Line 173ff is referring to citation 21. (Kawall 2020). However, I can’t find a supporting sentence in the referenced publication. At its best, Kawall 2020 hypothesizes on potential differences between genome editing and chemical/irradiation for induced mutagenesis but does not give concrete and convincing examples.

Second, Citation 37 does not provide an unambiguous assignment of a detection result to a specific product. In terms of specificity, the method does not meet the minimum requirements for qualitative test methods according to the currently valid MPR document. Therefore, a ‘softening’ of the wording is necessary:

Line 414: Alter “...detect the specific…” by “…detect a specific…”.

Line 458: substitute “shown” by “proposed” as citation 37 is still disputed.

Line 477: substitute “demonstrates” by “indicates” as citation 37 is still disputed

Line 479: substitute “can be” by “may be” as citation 37 is still disputed

Author Response

R…Reviewer’s comments

A…Authors’ reply

R: The manuscript has substanially improved. I appreciate that the authors considered most of my recommendations. However two important issues remain:

R: First, I strongly recommend deleting both citation 20 and 21. This step would not not alter the scientific contend of the manuscript.

Reasoning:

1a) Line 170ff: is referring to the following sentence in Berteau 2019 ““Recurrent detection of differences (on target and off-targets) and close PAM16,17 sequence [s] would form a strong predictor, not to say an unambiguous signature of the employed technique.” The ‘comprehensive’ review of Berteau 2019 (as a non-peer review book chapter) is disputed, e.g. by the German Biosafety Commission ZKBS 2020: “Bertheau suggests that unintentional and spontaneous mutations could be distinguished from intended alterations and claims that the applied New Breeding Technique could be identified as well, all by characterization of the organism, especially through phenotype characterization and different sequencing procedures. This suggestion lacks a scientifically sound explanation and is unfeasible according to the state-of-the-art of science and technology.” For more details: https://www.zkbs-online.de/ZKBS/EN/01_Aktuelles/ZKBS-commentary%20on%20Y.%20Bertheau%20(2019)/ZKBS-commentary%20on%20Y.%20Bertheau%20(2019)_node.html;

A: Thank you for pointing out that the text has to be improved for more clarity, as our intention was not to refer to the above-mentioned sentence or interpretation. In the cited book chapter Bertheau (2019) provides, in our opinion, a comprehensive overview of sequence changes, including their potential origin, that are potentially present in genomes that have been subject to genome editing. In the manuscript we do not evaluate to what extent and in which context these sequence alterations are useful for detection. Our reasoning includes that identification may be based on the analytical detection of sequence alterations that are present in a specific combination. As a consequence, there is no point in differentiating whether these alterations have been intended or not. It is more important that these alterations characterize a specific product. This is obviously hard or impossible to achieve if the sequence alteration does not go beyond a sole SNV.

To avoid misinterpretation, we have rephrased and merged sentences, and omitted the phrasing in brackets. For the above-mentioned reasons, we have, however, decided to keep the reference.

R: 1). Line 173ff is referring to citation 21. (Kawall 2020). However, I can’t find a supporting sentence in the referenced publication. At its best, Kawall 2020 hypothesizes on potential differences between genome editing and chemical/irradiation for induced mutagenesis but does not give concrete and convincing examples.

A: Thank you, also here we see the need for improvement to avoid misunderstanding. In the cited article of Kawall (2019) a chapter is dedicated to “Genetic variation through meiotic recombination”. In this chapter the author describes that the “probability of separating” plant genes that “have a low genetic distance” is “highly unlikely during breeding”. In our reasoning, this is an important point when considering the above-mentioned combinations of sequence alterations, as their exploitation for detection is particularly useful if they cannot be separated by crossing/backcrossing.

R: Second, Citation 37 does not provide an unambiguous assignment of a detection result to a specific product. In terms of specificity, the method does not meet the minimum requirements for qualitative test methods according to the currently valid MPR document. Therefore, a ‘softening’ of the wording is necessary:

Line 414: Alter “...detect the specific…” by “…detect a specific…”.

Line 458: substitute “shown” by “proposed” as citation 37 is still disputed.

Line 477: substitute “demonstrates” by “indicates” as citation 37 is still disputed

Line 479: substitute “can be” by “may be” as citation 37 is still disputed

A: Lines 414 and 479 have been amended as suggested by the Reviewer.

Line 458: We have decided to not follow the suggestion but have replaced “shown” by “published” to acknowledge the Reviewer’s comment that the publication is disputed.

Line 477: We acknowledge the Reviewer’s comment but prefer to change the wording to “illustrate”, which is used in the following sentence. To avoid word repetitions, we have merged the two sentences.

Authors’ remark: Please note that in addition to the Reviewer’s comments we have re-formatted the references where appropriate.